# A *qnrD*-Plasmid Promotes Biofilm Formation and Class 1 Integron Gene Cassette Rearrangements in *Escherichia coli*

**DOI:** 10.3390/antibiotics11060715

**Published:** 2022-05-26

**Authors:** Anamaria Babosan, Margaux Gaschet, Anaëlle Muggeo, Thomas Jové, David Skurnik, Marie-Cécile Ploy, Christophe de Champs, Fany Reffuveille, Thomas Guillard

**Affiliations:** 1Inserm UMR-S 1250 P3Cell, SFR CAP-Santé, Université de Reims-Champagne-Ardenne, 51100 Reims, France; anababosan@gmail.com (A.B.); amuggeo@chu-reims.fr (A.M.); cdechamps@chu-reims.fr (C.d.C.); 2Université de Limoges, Inserm, CHU Limoges, UMR-S 1092, 87032 Limoges, France; margaux.gaschet@unilim.fr (M.G.); thomas.jove@unilim.fr (T.J.); marie-cecile.ploy@unilim.fr (M.-C.P.); 3Laboratoire de Bactériologie-Virologie-Hygiène Hospitalière-Parasitologie-Mycologie, CHU Reims, Hôpital Robert Debré, Avenue du Général Koenig, CEDEX, 51092 Reims, France; 4Institut Necker-Enfants Malades, Inserm U1151-Equipe 11, Université Paris Descartes, 75015 Paris, France; david.skurnik@inserm.fr; 5Laboratoire de Bactériologie, AP-HP, Hôpital Necker-Enfants Malades, 75015 Paris, France; 6Division of Infectious Diseases, Department of Medicine, Brigham and Women’s Hospital, Harvard Medical School, Boston, MA 02115, USA; 7EA 4691 BiOS, SFR CAP-Santé, Université de Reims Champagne-Ardenne, 51100 Reims, France; fany.reffuveille@univ-reims.fr

**Keywords:** biofilm, *qnr*, *Escherichia coli*, integron

## Abstract

Bacteria within biofilms may be exposed to sub-minimum inhibitory concentrations (sub-MICs) of antibiotics. Cell-to-cell contact within biofilms facilitates horizontal gene transfers and favors induction of the SOS response. Altogether, it participates in the emergence of antibiotic resistance. Aminoglycosides at sub-MICs can induce the SOS response through NO accumulation in *E. coli* carrying the small plasmid with the quinolone resistance *qnrD* gene (pDIJ09-518a). In this study, we show that in *E. coli* pDIJ09-518a, the SOS response triggered by sub-MICs of aminoglycosides has important consequences, promoting genetic rearrangement in class 1 integrons and biofilm formation. We found that the integrase expression was increased in *E. coli* carrying pDIJ09-518a in the presence of tobramycin, which was not observed for the WT isogenic strain that did not carry the *qnrD*-plasmid. Moreover, we showed that biofilm production was significantly increased in *E. coli* WT/pDIJ09-518a compared to the WT strain. However, such a higher production was decreased when the Hmp-NO detoxification pathway was fully functional by overexpressing Hmp. Our results showing that a *qnrD*-plasmid can promote biofilm formation in *E. coli* and potentiate the acquisition and spread of resistance determinants for other antibiotics complicate the attempts to counteract antibiotic resistance and prevention of biofilm development even further. We anticipate that our findings emphasize the complex challenges that will impact the decisions about antibiotic stewardship, and other decisions related to retaining antibiotics as effective drugs and the development of new drugs.

## 1. Introduction

Misuse or overuse of antibiotics, in humans and in animals, is one of the main drivers of resistance. Anthropogenic pollution led to the release of pharmaceutical residues, including antibiotics, into the environment, and can accumulate in the sediments where bacteria are integrated into biofilms. Biofilms are also involved in infections related to medical devices. Cell-to-cell contact within biofilms facilitates horizontal gene transfers, participating in antibiotic resistance genes dissemination, and favors the induction of the SOS response. It has been shown that starvation, associated with the SOS response, mediates high biofilm-specific tolerance to the fluoroquinolone ofloxacin [1]. Moreover, in biofilms, the microbial community is surrounded by a protective extracellular matrix, which acts as a defense against all types of antimicrobials [2]. Due to higher Minimum Bactericidal Concentrations (MBC) as well as Minimum Inhibitory Concentrations (MIC), bacteria within biofilms may be exposed to sub-minimum inhibitory concentrations (sub-MICs) of antibiotics [3] that also participate in the induction of biofilm formation [3,4,5].

We recently showed that aminoglycosides at sub-MICs can induce the SOS response through NO accumulation in *E. coli* carrying small plasmids harboring the quinolone resistance *qnrD* gene, such as pDIJ09-518a-like plasmids [6]. The NO accumulation is due to higher NO formation and repression of the Hmp-mediated detoxification pathway, both driven by proteins encoded by the small *qnrD*-plasmid pDIJ09-518a. We showed that the gene encoding a putative FAD-binding oxidoreductase, ORF3, induces NO production with concomitant detoxification of NO hampered by the putative CRP/FNR-like protein encoded by ORF4, leading to inhibition of *hmp* expression. The NO accumulation and SOS induction are worrisome in terms of emergence of antibiotic resistances in *E. coli* carrying *qnrD*-plasmids. First, because some studies reported that NO has an effect on biofilm dispersal under aerobic conditions, but others pointed out that NO-mediated signals could promote biofilm formation in order to acquire a defense strategy against damaging agents or eukaryotic antimicrobial factors [7]. Secondly, in integrons, which are genetic elements involved in antibiotic resistance dissemination in Gram-negative bacteria [8,9], it has previously been reported that induction of the SOS response by fluoroquinolones or β-lactams promotes antibiotic resistance in *E. coli* by the class 1 integrons integrase-mediated incorporation and/or rearrangement of gene cassettes. 

In this study, we show that in *E. coli* harboring the small *qnrD*-plasmid, the SOS response triggered by sub-MICs of aminoglycosides has important consequences, promoting genetic rearrangement in class 1 integrons and biofilm formation. Overall, this finding complicates the attempts to counteract antibiotic resistance and prevention of biofilm development even further.

## 2. Results

### 2.1. Expression of Class 1 Integron Integrase Is Increased in qnrD-Plasmid-Carrying E. coli Exposed to Aminoglycosides

Fluoroquinolones and aminoglycosides are major classes of antibiotics used in medicine [10,11]. Therefore, it was all the more important to first evaluate class 1 integron to measure the potential impact of our new description of SOS induction upon aminoglycosides exposure. To investigate whether such SOS-dependent genetic rearrangements may also occur upon exposure to aminoglycosides, we used qRT-PCR to assess the expression of the class 1 integron integrase gene in isogenic *E. coli* MG1656 (WT) carrying (or not) the small *qnrD*-plasmid (WT/pDIJ09-518a and WT). As shown in Figure 1A, pDIJ09-518a did not increase the integrase expression in antibiotic-free LB medium. However, the integrase expression was increased in *E. coli* carrying the pDIJ09-518a in the presence of tobramycin. This result was not observed for the WT isogenic strain that did not carry pDIJ09-518a (Figure 1B).

Our finding that harboring the *qnrD*-plasmid induced the overexpression of the integrase of the class 1 integrons could have major clinical impact, as it indicates that exposure to aminoglycosides of these *E. coli* strains could therefore lead to both high-level expression of fluoroquinolone resistance and acquisition and spread of antibiotic resistance determinants through the class 1 integrons.

### 2.2. Small qnrD-Plasmid Enhances Biofilm Production in E. coli

In addition to directly promoting antibiotic resistance, we questioned if *qnrD*-plasmid carriage, which favors NO accumulation, could also give a selective advantage to *E. coli* to counter the pressure applied by antibiotics by enhancing biofilm production. Considering that NO accumulation resulted from NO production by ORF3 and NO detoxification inhibition by ORF4 [6], we studied biofilm formation using the strains *E. coli* WT and WT/pDIJ09-518a, as well as the deleted derivatives of ORF3 and/or ORF4; WT/pDIJ09-518aΔORF3, WT/pDIJ09-518aΔorf4), and WT/pDIJ09-518aΔORF3ΔORF4, respectively.

We first quantified the planktonic and biofilm cell concentrations using a crystal violet staining assay for WT *E. coli* and *E. coli*/pDIJ09-518a. We observed that biofilm production was significantly increased in *E. coli* WT/pDIJ09-518a compared to the WT strain (Figure 2A). With the strains deleted of ORF3 and/or ORF4, we observed a trend (*p* = 0.6 for the double mutant ΔORF3ΔORF4) for less biofilm formation compared with the WT/pDIJ09-518a strain.

Next, we took a very sensitive approach to determine biofilm production in a dynamic setting, using a biofilm flow cell assay to measure thickness, overall structure, and the total percentage of dead or live cells in biofilms (Figure 2B) [12]. In this biofilm model, the number of live cells increased for the WT/pDIJ09-518a strain compared to the WT *E. coli*. When the Hmp-NO detoxification pathway was fully functional, either by Hmp overexpression (WT/pDIJ09-518a/pHmp) or by ORF3-ORF4 deletion, much less biofilm was formed (Figure 2B). As shown in the three-dimensional pictures (Figure 2C), WT/pDIJ09-518a biofilm was slightly taller (35 µm vs. 25 µm) but with a much denser structure than the WT biofilm, indicating the increased capacity of live *E. coli*/pDIJ09-518a cells to form a network of strong, well-organized micro-colonies consistent with a mature and drug-resistant biofilm.

## 3. Discussion

Fluoroquinolones, aminoglycosides, and β-lactams are the three classes of antibiotics most often used in medicine [10,11]. Since the advent of effective antimicrobial chemotherapy, mortality from infectious diseases has decreased significantly, but this has also led to an increased number of drug-resistant bacteria that threaten the lifesaving capabilities of these essential drugs. Selective pressures maintained by overuse and misuse of antibiotics by humans is the main driver of resistance, but antibiotic use in animals and accumulation in the environment also contribute to this problem. In addition, when low concentrations of antibiotics are present that are unable to kill bacteria (sub-MIC), they can select for resistance and tolerance through induction of biofilm development [3,4,5], and this occurs notably with fluoroquinolones [13]. 

We recently reported the induction of the SOS response in *E. coli* upon exposure to sub-MICs of aminoglycosides [6]. Here, we show two additional consequences (enhanced class 1 integrase expression and biofilm formation) from acquiring the small transmissible *qnrD*-plasmids, which lead *E. coli* to increase its ability to persist in humans and in the environment (Figure 3).

In this study, we showed that the class 1 integron integrase is more highly expressed in the presence of sub-MIC of tobramycin in WT/pDIJ09-518a. This increased expression can induce recombination events, leading to acquisition or rearrangements of antibiotic resistance gene cassettes. Integrase-mediated rearrangements can generate integron variants in which a weakly expressed gene cassette moves closer to the gene cassette promoter, thus leading to higher-level resistance. Moreover, the integrase expression has been shown to be increased in biofilms through both SOS and biofilm-specific regulations of the integrase [14]. Strains co-carrying *qnrD* and other antibiotic resistance determinants embedded in class 1 integrons, such as extended-spectrum β-lactamases or carbapenemases, have been described [15,16]. This raises the concern that in *E. coli* isolates co-harboring small *qnrD*-plasmids, the spread of integron-mediated antibiotic multi-drug resistance could be induced by sub-MICs of aminoglycosides, leading not only to high-level fluoroquinolone resistance, but also resistance to last-resort antibiotics such as carbapenems.

We also showed that biofilm formation is enhanced in *E. coli* carrying the *qnrD*-plasmid pDIJ09-518a. This could be of major concern in infections related to medical devices usually associated with biofilms. Indeed, antibiotic treatments classically fail to reach optimal concentration in biofilms. In such infections caused by *E. coli* carrying small *qnrD*-plasmids, aminoglycosides treatments may achieve only sub-lethal concentrations at the infection site, leading to cross-selection for resistance to fluoroquinolones and induction of integron-mediated gene cassette rearrangements. 

Overall, our findings revealed a key element in the role of environmental factors in antibiotic resistance. Fluoroquinolones and aminoglycosides are poorly biodegradable and accumulate in wastewaters [17]. Therefore, given consumption of these antibiotics in hospitals and the presence of *qnrD* in sewage water [18,19], wastewaters could be potent reservoirs for selection of fluoroquinolone-resistant bacteria, promoting horizontal gene transfer from environmental bacteria to pathogens and having consequences for their capacity to form biofilm.

Such mobilizable small *qnrD*-plasmids conferring low-level resistance to fluoroquinolones have been described in several enterobacterial species, and more specifically in *Morganellaceae* [20,21]. Therefore, our findings showing that a *qnrD*-plasmid can promote biofilm formation in *E. coli* and induction of the SOS response by aminoglycosides, with the consequence of potentiating the acquisition and spread of resistance determinants for other antibiotics, are worrisome. This emphasizes the complex challenges that will impact the decisions about antibiotic stewardship, other decisions related to retaining antibiotics as effective drugs, and development of new drugs. Our findings might be a harbinger of additional and complex interactions among drugs and bugs that will impact how the medical community proceeds in its future decisions about deploying existing and newer antimicrobial chemotherapeutic agents.

## 4. Material and Methods

### 4.1. Bacterial Strains, Plasmids, Primers, and Growth Conditions

The bacterial strains, plasmid constructs, and primers for PCR analysis are shown in Table 1. Experiments were performed in LB medium or in minimum medium at 37 °C. For genetic selections, antibiotics were added to the media at the following concentrations: ciprofloxacin 0.06 μg /mL and kanamycin 50 μg/mL.

### 4.2. DNA Manipulation and Genetic Techniques

Genomic DNA (gDNA) was extracted and purified using the Qiagen DNeasy purification kit (Qiagen, Courtaboeuf, France). Isolation of plasmid DNA was carried out using the QIAprep Spin Miniprep kit (Qiagen). Gel extractions and purifications of PCR products were performed using the QIAquick Gel Extraction kit (Qiagen) and QIAquick PCR Purification kit (Qiagen). PCR verifying experiments were performed with Go Taq Green Master Mix (Promega, Charbonnières les Bains, France), and PCRs requiring proofreading were performed with the Q5^®^ High-Fidelity DNA Polymerase (New England BioLabs, Evry, France) as described by the manufacturers. Restriction endonucleases DpnI were used per the manufacturer’s specifications (New England BioLabs). All DNA manipulations were checked by DNA sequencing carried out by GENEWIZ Europe (Takeley, England).

### 4.3. Plasmid Constructions

The *hmp* gene with its own promoter was amplified from the *E. coli* MG1656 genome, with the corresponding Forward/Reverse primers shown in Table 2. The PCR products were purified and cloned into pCR2.1^®^ (ThermoFisher Scientific, Illkirch-Graffenstaden, France) to generate pHmp and selected on plates containing 50 µg kanamycin/mL. 

pDIJ09-518aΔORF3, -ΔORF4, and -ΔORF3ΔORF4 were obtained by PCR amplification of a 5′ and 3′ fragment of the ORF3- and ORF4-encoding genes, using the native pDIJ09-518a plasmid as DNA template and the primers described in Table 2. The primers were obtained using the NEBuilder Assembly Tool (New England Biolabs). After digestion by DpnI (New England Biolabs) and purification of PCR products (Qiagen), the fragments obtained were transformed into electrocompetent *E. coli* WT (MG1656). Transformants were selected on agar plates containing 0.06 µg ciprofloxacin/mL and were analyzed by PCR as described above. 

### 4.4. RNA Extraction and qRT-PCR

Strains were grown in LB at 37 °C, with shaking, to exponential phase (OD_600_ = 0.6). Six biological replicates were prepared. One percent of the MIC of tobramycin (MIC = 0.125 μg/mL for all isogenic derivative strains from *E. coli* WT) was then added to the culture for 30 min to allow the induction of the SOS response. One culture was kept as an antibiotic-free control. RNA of 1.5 mL of exponentially growing cells was extracted using Nucleospin RNA (Macherey Nagel, Düren, Germany). The gDNA contaminating the samples was removed with TURBO DNA-free Kit (Ambion, ThermoFisher Scientifics) at 37 °C, for 30 min. First-strand cDNA synthesis was performed with five hundred nanograms of treated RNA with PrimeScript RT Reagent Kit (Takara Bio Inc., Shiga, Japan). Quantitative real-time PCR were performed using the SYBR^®^Green FastMix (Avantor, VWR, Rosny-sous-Bois, France) on the CFX96 (BioRad, Hercules, CA, USA) using the primers indicated in Table 2. We assessed *intl1* expression in strains grown with tobramycin compared to strains grown without antibiotics by performing an absolute quantification of *intl1* and the endogenous gene *dxs*. 

### 4.5. Biofilm Formation Assay

PVC 48-well microtiter plates (Corning, Saint-Quentin-Fallavier, France) were used to monitor biofilm formation as described previously [22]. Briefly, minimal media were inoculated with a 1/100 dilution from an overnight culture in LB media. After inoculation, microtiter plates were incubated at 37 °C for 24 h and rinsed. The OD_600_ of the supernatant was determined (planktonic growth). Then, 500 μL of a 0.1% solution of crystal violet was added to each well. The plates were incubated at room temperature for 20 min and rinsed. Biofilm formation was tested as follows: crystal violet was solubilized by addition of 200 μL of 95% ethanol and the OD_595_ was determined. Results are presented as the mean of four replicates.

### 4.6. Dynamic Biofilm Model

The biofilms were established as previously described [23]. They were grown for 24 h at 37 °C in flow chambers with channel dimensions of 1 by 4 by 40 mm. Briefly, the system was assembled and sterilized by pumping a 0.5% hypochlorite solution and rinsed with sterile water and medium. After injection of 400 µL of an overnight culture diluted to an OD_600_ of 0.05, chambers were left without flow for 2 h. The culture medium was then pumped through the system at a constant rate of 2 mL/h for 24 h. Biofilms were stained using the LIVE/DEAD BacLight Bacterial Viability kit (Molecular Probes, Eugene, OR, USA) or SYTO-9 alone prior to microscopy experiments. A ratio of SYTO-9 (green fluorescence, live cells) to propidium iodide (PI) (red fluorescence, dead cells) of 1:5 was used. Microscopy was performed using a confocal laser-scanning microscope (LSM 710 NLO, ZEISS, Jena, Germany) and three-dimensional reconstructions were generated using the Imaris software package (Bitplane AG). Biofilm surface (µm^3^) was calculated using Imaris software.

### 4.7. Statistics Analysis

For RT-qPCR, a Wilcoxon matched-pairs signed-rank test was used to compare the median of fold changes [24]. Data represent median values of 6 independent biological replicates, and error bars indicate upper/lower values. * *p* < 0.05 Wilcoxon matched-pairs signed-rank test. For biofilm dynamic assay formation, data from confocal microscopy imaging with ALIVE/DEAD bacteria within the biofilm were analyzed using a 2-way ANOVA with a *p* value < 0.05 for strains as a source of variation in the overall ANOVA and a * *p* < 0.05 using Dunn’s multiple comparisons test. All the tests were performed using GraphPad Prism version 7.

**Table 1 antibiotics-11-00715-t001:** Strains and plasmids.

Strains	Genotype/Description	References/Sources
MG1656 (WT)	Δ*lacI-lacZ* derivative of MG1655	(*50*)
WT/pDIJ09-518a	MG1656 carrying pDIJ09-518a, Cip^R^	[6]
WT/pDIJ09-518aΔORF3	MG1656 carrying pDIJ09-518a deleted for ORF3, Cip^R^	[6]
WT/pDIJ09-518aΔORF4	MG1656 carrying pDIJ09-518a deleted for ORF4, Cip^R^	[6]
WT/pDIJ09-518aΔORF3ΔORF4	MG1656 carrying pDIJ09-518a deleted for ORF3 and ORF4, Cip^R^	[6]
WT/pDIJ09-518a/pHmp	MG1656 carrying pDIJ09-518a and plasmid over-expressing Hmp protein	[6]
TOP10	Transformation strain	Invitrogen
**Plasmids**		
pDIJ09-518a	CIP^R^	Lab collection
pDIJ09-518aΔORF3	pDIJ09-518a deleted for ORF3, Cip^R^	[6]
pDIJ09-518aΔORF4	pDIJ09-518a deleted for ORF4, Cip^R^	[6]
pDIJ09-518aΔORF3ΔORF4	pDIJ09-518a deleted for ORF3 *and* ORF4, Cip^R^	[6]
pHmp	pTOPO::Hmp, Km^R^	[6]
p1W	pSU38 derivative plasmid containing a complete class 1 integron (*intI1* with a PcW promoter), Km^R^	[25]
pZA2	pZA2 plasmid, Km^R^	[26]
pZA2_intI1PcW_noK7	pZA2 derivative carrying *intI1* (PcW promoter) under control of its native promoter. *intI* amplified from p1W with primers YL1 and YL2 and cloned into the XhoI and BamHI restriction sites.	[27]

**Table 2 antibiotics-11-00715-t002:** Primers used for this study.

Primers	Sequence (5′-3′)	
TG01	GGAGCTGATTTTCGAGGG	To check *qnrD* by sequencing
TG02	AGAAAAATTAGCGTAACTAAGATTTGTC	To check *qnrD* by sequencing
LC3	ATGACGTGGCGATTCAAAA	To amplify *dxs*
LC4	AGCCGGTATAGAGCATCTGG	To amplify *dxs*
AB01	GTTGTCTATCGCGAAGATCAG	To amplify *sfiA*
AB02	GAGCTGGCTAATCTGCATTAC	To amplify *sfiA*
TG08	CATCCGCATCTCCTGACTCA	To amplify *hmp* and its own promoter
TG09	GCGCAAACCGGCAAAATCG	To amplify *hmp* and its own promoter
TG10	GTAAAACGACGGCCAGT	To check insert cloned in pTOPO by sequencing
TG11	CAGGAAACAGCTATGAC	To check insert cloned in pTOPO by sequencing
AB21	TACTGTCTCCGTTCACACATGATCGGAGGGTGTCTCCGTTAGGTTTAC	To allow ORF3 deletion
AB22	GAGACACCCTCCGATCATGTGTGAACGGAG	To allow ORF3 deletion
AB25	TTGCACCCCATGATACTTTCAGTATCCTTTCAGCGATTTC	To allow ORF4 deletion
AB26	GATACTGAAAGTATCATGGGGTGCAA	To allow ORF4 deletion
AB29	TACTGTCTCCGTTCACACATGATCGGAGGGTGTCTCCGTTAGGTTTAC	To allow ORF3 and ORF4 deletion
AB30	GATACTGAAAGTATCATGGGGTGCAA	To allow ORF3 and ORF4 deletion
YL1	CCGGAATTCTCGAGTACCTCTCACTAGTGAG	To amplify *intI1* with its promoter region
YL2	CTCTAGAGGATCCATACCTAACTTTGTTTTAGGGCGAC	To amplify *intI1* with its promoter region

## Figures and Tables

**Figure 1 antibiotics-11-00715-f001:**
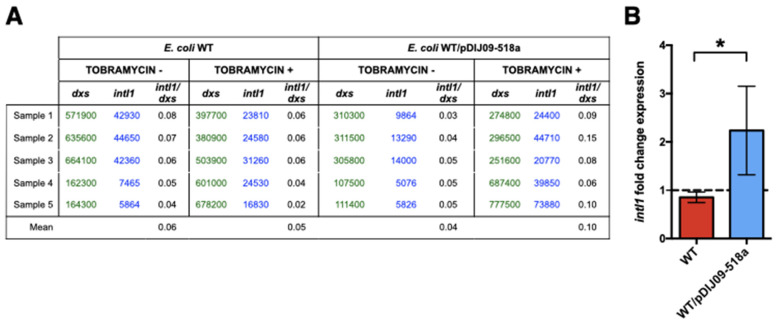
Integrase of class 1 integrons is overexpressed in *E. coli* harboring the small *qnrD*-plasmid upon exposure to tobramycin. (**A**) Copy numbers of *intl1* determined using absolute quantification of *intl1* and *dxs*, in *E. coli* MG1656 (WT) and its derivative carrying pDIJ09-518a (WT/pDIJ09-518a); (**B**) fold-change expression of *intl1* in WT and WT/pDIJ09-518a exposed to tobramycin in comparison to LB. Data represent median values of 6 independent biological replicates, and error bars indicate mean with SD. * *p* < 0.05 Mann–Whitney test.

**Figure 2 antibiotics-11-00715-f002:**
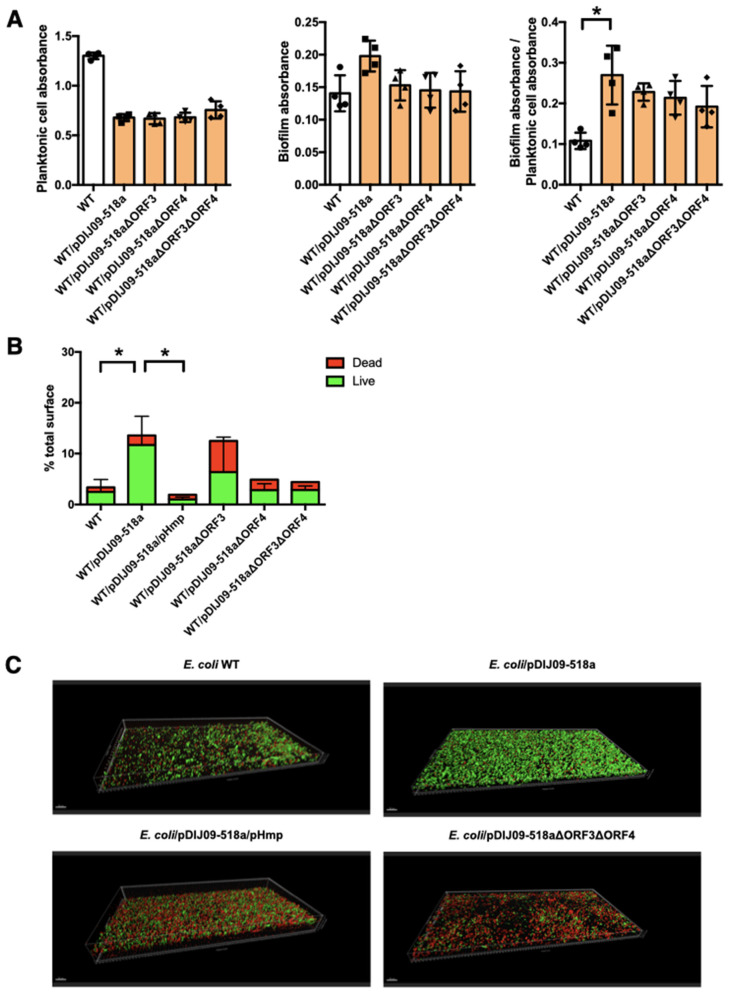
Aminoglycosides promote biofilm formation in *E. coli* carrying the small *qnrD*-plasmid. (**A**) Crystal violet binding assay for assessment of biofilm formation by *E. coli* WT (white) and its derivatives strains (orange): WT/pDIJ09-518a (carrying the native plasmid, square) WT/pDIJ09-518aΔORF3 (carrying the plasmid deleted for ORF3, upward triangle), WT/pDIJ09-518aΔORF4 (WT/pDIJ09-518a Δorf4, downward triangle) and WT/pDIJ09-518a ΔORF3ΔORF4 (carrying the plasmid deleted for both ORF3 and ORF4, diamond). Left panel, crystal violet planktonic absorbance (OD_600_). Middle panel—crystal violet biofilm absorbance (OD_595_). Right panel—biofilm index (crystal violet absorbance/planktonic cell absorbance (OD_595_/OD_600_). Data were analyzed using a 2-way ANOVA with a *p* value < 0.05 for strains as a source of variation in the overall ANOVA. * *p* < 0.05 using Dunn’s multiple comparisons test. Mean rank differences for WT compared to WT/pDIJ09-518a and WT/pDIJ09-518aΔorf3 were −13 and −10.5, respectively. Error bars represent the SD. (**B**,**C**) Confocal microscopy imaging for assessment of three-dimensional biofilm formation by *E. coli* WT, WT/pDIJ09-518a, WT/pDIJ09-518a/pHmp, WT/pDIJ09-518aΔORF3, WT/pDIJ09-518aΔORF4, and WT/pDIJ09-518a ΔORF3ΔORF4. The alive bacteria embedded in the biofilm are depicted in green, while the dead ones are depicted in red. Data were analyzed using a 2-way ANOVA with a *p* value < 0.05 for strains as a source of variation in the overall ANOVA. * *p* < 0.05 using Dunn’s multiple comparisons test. Mean rank differences for WT compared to WT/pDIJ09-518a was −10 and 14 for WT/pDIJ09-518a compared to WT/pDIJ09-518a/pHmp. Error bars represent the SD.

**Figure 3 antibiotics-11-00715-f003:**
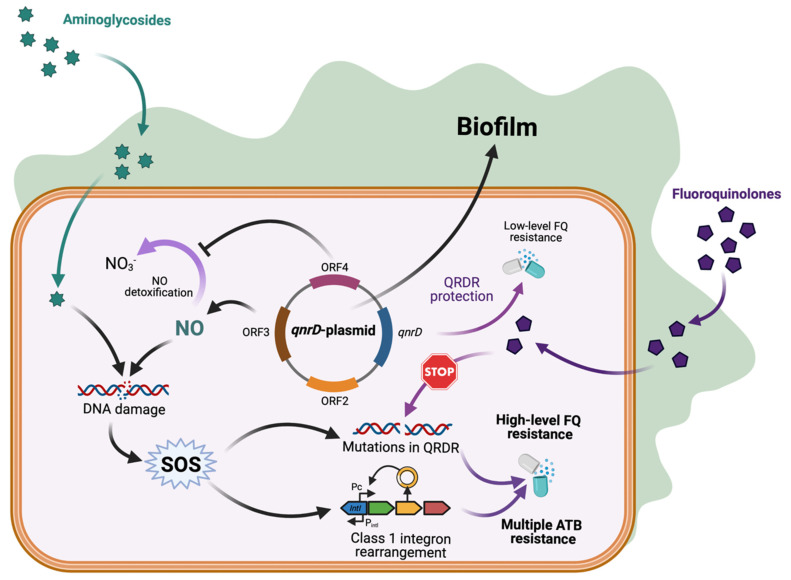
*qnrD*-plasmid promotes biofilm formation and class 1 integron rearrangement with antibiotic resistance consequences. Enhanced biofilm formation by *qnrD*-plasmid favors exposure to sub-MICs. For fluoroquinolones it increases SOS-mediated *qnrD* expression and low-level of fluoroquinolones resistance. For aminoglycosides, the NO accumulation-mediated SOS induction triggers mutagenic response and class 1 integron rearrangement, both leading to potential multiple antibiotic resistance. Created with BioRender.com (accessed on 10 May 2022).

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
