# Peer review of "A qnrD-Plasmid Promotes Biofilm Formation and Class 1 Integron Gene Cassette Rearrangements in Escherichia coli"

_antibiotics, 2022, doi:10.3390/antibiotics11060715_

Round 1

Reviewer 1 Report

In this research paper, authors have studied Biofilm Formation and Class 1 Integron Gene Cassette Rearrangements in Escherichia coli. They have overexpressed class 1 integrons in wild type E. coli harboring the small qnrD-plasmid to check the expression of class 1 integron integrase and biofilm production. This research paper useful for the researcher to utilize this approach to study the integrase overexpression and biofilm production in E. coli. Therefore, this reviewer would like to suggest that the manuscript is acceptable for publication in the journal after minor revision

The following changes should be made throughout the manuscript

This reviewer suggests changing the title of article:

Overexpression of qnrD Gene Promotes the Biofilm Formation and Class 1 Integron Gene Cassette Rearrangements in Escherichia coli

Antibiotics concentration should be written in following format:

ciprofloxacin (0.06 μg/mL) and kanamycin (50 μg/mL)

Author Response

Please find in the attached our reply to your comments.

Reviewer 2 Report

Journal: antibiotics

Manuscript ID: antibiotics-1744645

Title:

“A qnrD-plasmid promotes biofilm formation and class 1 integron gene cassette rearrangements in Escherichia coli”

Authors:

Anamaria Babosan , Margaux Gaschet , Anaëlle Muggeo , Thomas Jové , David Skurnik , Marie-Cécile Ploy , Christophe De Champs , Fany Reffuveille , Thomas Guillard

Authors examine the role of a qnrD-plasmid in the promotion of biofilm formation in E. coli ad in the acquisition and spread of resistance determinants for other antibiotics. The manuscript is well organized, easy to read and the experimental methods and results well presented. I only have some minor changes/questions.

  1. Line 22: “…. qnrD gene (pDIJ09-518a).In this…..”

A space is missing

  1. Line 29: “….l by overexpressing Hmp.Our….”

A space is missing

  1. In RT-qPCR experiments the fold change expression was calculated with the ∆∆Ct method? This should be mentioned in the manuscript
  2. Was the amplification efficiency (E) calculated for intl1? This is a very important value for the RT-qPCR experiments, as a poor efficiency puts the Rt-qPCR results in doubt.
  3. For the RT-qPCR statistical analysis, why was a non-parametric method used, as parametric methods are usually preferred? Usually with a n=9 the RT-qPCR experiments give good results for narmality. Did you check the normality of your values before applying the non-parametric test?

Author Response

Please find in the attached file our reply to your comments
